# Barriers and facilitators for the provision of radiology services in Zimbabwe: A qualitative study based on staff experiences and observations

**Saba Hinrichs-Krapels**[1‡], **Lazarus Tombo**[2‡], **Harriet Boulding**[3], **Edith D. Majonga**[4], **Carole Cummins**[5], **Semira Manaseki-Holland**[5]*

1 Faculty of Technology, Policy and Management, Delft University of Technology, Delft, The Netherlands, 2 Radiology Department, University Hospitals Birmingham NHS Foundation Trust, Birmingham, United Kingdom, 3 The Policy Institute, King's College London, London, United Kingdom, 4 Department of Medical Physics and Imaging Sciences, University of Zimbabwe College of Health Sciences, Harare, Zimbabwe, 5 Institute of Applied Health Research, University of Birmingham, Birmingham, United Kingdom

‡ SHK and LT contributed equally to this work as joint first authors.
* s.manasekiholland@bham.ac.uk

**Data Availability Statement:** Due to ethical concerns the researchers are unable to provide the data within your submission. For more information

## Abstract

Despite its fundamental role in diagnostic and curative care, radiology has been described as a neglected essential service in many low and middle-income countries (LMICs). Previous studies have demonstrated basic equipment and infrastructure shortages in LMIC settings, but no studies to date have gone further in understanding the perceptions and experiences of staff delivering radiology services, as a way of identifying their perspectives on barriers and facilitators for delivering services, and the potential for where improvements can be made. Our qualitative study aimed to: (a) identify barriers for delivering radiology services, and (b) suggest potential facilitators for improvement of radiology service delivery in the Zimbabwean context; from the perspective of radiology staff. We conducted semi-structured interviews (n = 13) and three focus groups (n = 24 radiographers), followed by four half- to full- days of field observations to validate insights from the interviews and focus groups in all three public hospitals and one private hospital in the Harare metropolitan area. Our study identified four main barriers for delivering radiology services: (i) poor basic infrastructure, equipment, and consumables; (ii) suboptimal equipment maintenance; (iii) shortage of radiology staff and skills development; and (iv) lack of wider integration and support for radiology services. We also identified a strong sense of motivation among staff to keep radiology services, pointing to what may be an enabler and facilitator for improving radiology services. These findings point to potential risks to patient safety and quality of delivering radiology services. More importantly, we found a strong sense of personal motivation displayed by the staff, suggesting there is the potential to maintain and improve existing practices, but this would require investments to train and remunerate more radiology staff, as well as investing in continuing professional development.

please contact the ethical body mandating this restriction at the University of Birmingham, researchgovernance@contacts.bham.ac.uk.

**Funding:** The authors received no specific funding for this work.

**Competing interests:** The authors have declared that no competing interests exist.

## Introduction

According to the World Health Organisation (WHO), diagnostic imaging and radiology services are crucial for "confirming, correctly assessing and documenting courses of many diseases as well as in assessing responses to treatment" [1]. However, while radiology is fundamental for clinical diagnostics as well as curative services, it has been identified as a 'neglected essential service' in many low and middle-income country (LMIC) settings [2]. Previous studies of various aspects of radiology service delivery have highlighted issues such as the scarcity of radiology resources and units in Zimbabwe [3] and Tanzania [4], inadequately filled radiology forms in Ghana [5] and Nigeria [6], and the challenges and opportunities for implementing better quality services in Nepal [7], Guyana [8], and Rwanda [9]. As a result, there have been many efforts to address the shortages of imaging services to improve healthcare quality and increase health care disparities, including economic development for radiology services, public health mechanisms and education and training to optimise service quality [10].

Beyond these examples of studies, we have found no studies to date that have gone further in understanding the perceptions and experiences of staff delivering radiology services, beyond identifying shortages in equipment in an LMIC setting. Furthermore, we found no studies to date that specifically sought to identify both barriers and facilitators to providing better radiology service provision from the perspective of radiology staff. Including their perspective allows for a deeper understanding of how such infrastructure and equipment challenges provide a barrier to their work, and what insights and experiences of staff can lead to potential facilitators for improvement. By radiology services we refer to all activities that use medical imaging to diagnose and treat disease, including X-ray radiography, interventional radiology, ultrasound, computer tomography (CT), nuclear medicine and magnetic resonance imaging (MRI).

We examined these perceptions and experiences in the context of Zimbabwe, a low-income country in southern Africa with largely inadequate provision of healthcare. The country's healthcare system has been deeply affected by the economic crisis affecting the country since 1998 [11]. At the time of our study, Zimbabwe spent about 6% of its GDP on healthcare [12]. There is a critical shortage of health workers, with only 0.2 physicians and 1.9 nurses and midwives per 1,000 people [12]. According to the latest 2016 health access and quality index, Zimbabwe came 174th out of 195 countries in terms of access to personal health-care and quality of care [13]. Zimbabwe has eight provinces and two cities with provincial status, Harare (the capital) and Bulawayo. An audit conducted in 2019 identified 64 district hospitals in the country, most of which were government-run [3]. In the public healthcare sector, primary care facilities are the most accessed via 1331 health centres (according to the same audit), and most communities are within 8 kilometres of such a facility [3].

Radiology services in Zimbabwe are provided mainly in hospitals based in cities, and mostly in Harare. According to the aforementioned 2019 audit of licensed radiology equipment resources in Zimbabwe, public-sector access to X-ray units was 11 per 10 million people, which is approximately half the WHO recommendation of 20 for 10 million [3]. The same audit also found inequitable distribution of radiology resources: more than half of Zimbabwe's radiology equipment (215 out of 380) were in the two major cities of Harare and Bulawayo, serving one-fifth of the population, and two-thirds of all units (243 out of 380) were in the private sector hospitals which are routinely accessible by only 10% of the population [3]. This can disproportionally concentrate referrals to these regions and yet further widen the inequitable access to radiology services in the rest of the country. As suggested by the Maboreke et al. (2019), the national deficit in equipment can serve as a "proxy estimate of the additional

human resources, by way of radiographers, radiologists, and medical physicists required to coordinate a more equitable public-sector radiological service" [3].

Our qualitative study aimed to: (a) identify barriers for delivering radiology services, and (b) suggest potential facilitators for improvement of radiology service delivery in the Zimbabwean context; from the perspective of radiology staff. These aims were achieved via qualitative methods as outlined below. By reflecting on our findings and referring to existing literature on enabling quality improvement for radiology services, we discuss potential facilitators for improvement of radiology service delivery in the Zimbabwean context.

## Methods

### Ethics statement

Ethics approvals for the study were received from University of Birmingham ethics committee board (ERN_18–0237), Joint Research Ethics Committee for the University of Zimbabwe (JREC/209/18), Medical Research Council of Zimbabwe (MRCZ/B/1570). Entry permissions to conduct the study were granted by each participating hospital. A letter of invitation outlining aims of the project and extent of required participation was sent to the departmental managers which was then conveyed to the then potential participants, who were gathered with the assistance of departmental managers. The researcher (LT) explained the project at a general information meeting. Following this information provided at the meeting and the project information given in writing as part of the informed consent form, all participants who volunteered for the interviews and focus groups signed an informed consent form.

### Setting

Our study was conducted in the capital Harare, which represents the region with the highest concentration of radiology services as noted above and in a recent 2019 audit of radiology services [3].

### Study design and approach

Given the lack of existing empirical studies in the region relating to radiology services, our study was designed to be exploratory and qualitative, with the intention of understanding how staff experience delivering radiology services [14, 15], and exploring how their barriers to delivering their services could be overcome. We took mainly an inductive approach, based on empirical data collected, while guided in our data collection and analysis by current guidelines and standards for providing high quality radiology services from published academic literature and authoritative sources [16–24] (summarised in Table A1 in S1 Appendix). While we stress that our study was not designed to conduct a point-by-point comparison between these standards and current practice in Zimbabwe, we referred implicitly to these standards in three ways: (i) Interview questions (detailed below and provided in S2 Appendix) were guided by the factors affecting services mentioned in these standards, such as providing timely service, producing easy to read images, etc. (ii) When analysing the data (detailed below, in terms of interviews, focus groups, and observations) we identified any mentions that were related to these factors, and (iii) The main interviewer/observer was a radiologist whose perspective would inevitably have been influenced by his professional understanding of these standards.

### Sampling and data collection

Data was collected between October and November 2018 in all 3 available public (labelled Hospitals A, B and C in Table 1) and 1 private hospital (Hospital D), all in the Harare

**Table 1. Details of data collection methods (and participants where appropriate).**

| Method | Details |
|---|---|
| **Semi-structured interviews (n = 13)** | radiology managers (n = 4) |
| | radiographer (n = 1) |
| | senior radiographers (n = 4) |
| | radiologists (n = 2) |
| | consultant physician (n = 2) |
| **Focus group participants (n = 24) from three focus groups held at respective hospitals** | Radiographers, Hospital A (n = 9) |
| | Radiographers, Hospital B (n = 8) |
| | Radiographers, Hospitals C&D (n = 7) |
| **Observations** | Hospital A (1 day) |
| | Hospital B (2 days) |
| | Hospital C (1 day) |

Note: Given the small number of hospitals providing radiology services in Harare, we have not specified the location of each interviewee to avoid identification of participants.

metropolitan area. This sample represented a significant share of the total number of hospitals in the region, since at the time of our study there were a total of four public and two private hospitals that offered radiology services in the city. Given our emphasis on seeking the perspective of radiology staff for delivering quality services, our main method for data collection comprised of **semi-structured interviews** (n = 13 participants from a range of roles and hospitals) and three **focus groups** (n = 24 radiographers from the four hospitals). To provide further contextual information on the realities of delivering radiology services, and improve the validity [15] of data collected from the responses from the staff, we conducted four half to full days of **field observations** at the three public hospitals, and not the private hospital (Table 1).

We adopted a semi-structured interview approach, in which questions are open-ended, allowing the participants to comment around the question themes and provide in-depth data arising from the interaction [14]. A variety of staff involved in delivery of radiology services were included as interviewees to ensure breadth of focus and perspectives, recruited through self-selection in a convenience sample in each hospital. Despite resource constraints and the limited availability of staff for research recruitment, we noted saturation in the emerging themes in the responses during the interviews and focus groups.

The interview protocol is included In Annex B. Focus groups were used to draw upon participants' attitudes, perceptions and experiences in delivering radiology services, generating deeper and more nuanced understanding of their experiences [25]. These were conducted specifically with radiographers in the absence of managers, radiologist and physicians to avoid power differentials, reprisal and potential biases to arise. These discussions also enabled us to gain a large amount of information in a short period of time. Field observations in three hospitals were conducted to gain an impression of how the radiographers conducted their work in the radiology departments, and how they related to the patient in real settings; thereby providing challenge and/or validation of what was shared in the focus groups and interviews [26]. All interviews, focus groups and observations were conducted in English by LT. Audio recording was used and at the end of each day they were transcribed. Notes were taken during the focus groups and field observations by LT. Interviews were conducted in a quiet office within the hospital. Focus groups were held in conference rooms within the radiology department at three hospital sites.

## Analysis

Thematic analysis was used, which entailed manually identifying themes and patterns in qualitative data. Themes were first derived by the same researcher who collected the data (LT) and then repeated independently by HB and SHK for validation of emergent themes. During analysis, triangulation of the data collected through the different methods of semi-structured interviews, focus groups and observations were reviewed to validate the conclusions made from the data [27].

## Reflexivity

LT was the sole data collector, a UK qualified radiographer originally from Zimbabwe and had no direct relationship with the participants. His experience in the field and of radiology services meant he understood the challenges and requirements of delivering the services expressed by the study participants. However, we acknowledge that being trained in the UK can also introduce biases as to the expected standards of service and care, despite understanding the Zimbabwean culture. To mitigate this, we chose to compare any observations of current practice to global standards for radiology services, rather than expected standards from any one professional. The focus group and interview transcripts were analysed independently first by LT and then also by HB and SHK.

# Results

We identified barriers for delivering radiology services across four main themes: (i) basic infrastructure, equipment, and consumables (ii) equipment repair and maintenance (iii) radiology staff shortages and skills development, and (iv) wider integration and support. Furthermore, our analysis also identified a (v) strong sense of motivation among staff to keep radiology services, pointing to what may be an enabler and facilitator for improving radiology services, leading to our reflections in the discussion section.

## (i) Poor basic infrastructure, equipment, and consumables

Both field observations and interviews highlighted the scarcity of resources and basic infrastructure for delivering radiology services in the studied hospitals. These were specifically observed with respect to the building and radiology departments layouts, and the availability of equipment and consumables.

Field observations in all four hospitals revealed that the building designs did not meet basic radiology department requirements. In one hospital, the radiology department was on the first floor with no wheelchair access. Waiting areas in all hospitals were small with patients sitting on benches in narrow corridors, while some were forced to stand for extended periods of time due to lack of chairs or benches. The rooms were large enough for patients and care givers to provide their services, but they were not equipped with wall mounted oxygen or oxygen tanks present in the rooms. There were also often water, toilet paper and clothing equipment shortages.

Equipment availability was one of the issues that were raised by all interviewees and focus group participants, affecting the entire radiology service. Out of the four hospitals only two had MRI scanners, one in a private and one in a public hospital which was still relatively new at the time of observations (about 4 years old), and not yet fully in use.

None of the hospitals had interventional radiology or properly functioning fluoroscopy units. For those with the fluoroscopy rooms, the machines have not been working for years and the rooms were used as general storage rooms.

This lack of availability of equipment has very direct effects on patient care, as in the following example on children's services:

*"I would say the equipment that we have is not enough because we have children hospital, but we do not have the x-ray machines for the children department so instead the children have to move about 100 meters all the way from the children hospital to the adult hospital. Sometimes they even fail to do that because some of the departments don't have oxygen tanks so at the end of the day children go without radiological services . . .".*

[Radiographer 6]

Interviewees referred to certain brands of radiology equipment that were purchased due to lower costs, but these models were difficult to manoeuvre, and proved to be a hindrance to patient care:

*"[The equipment] not compatible with the trauma patient we are talking about. Of course, it is something . . . when you want a floating tube to do the HTP (horizontal tube projection) it is impossible and. . . you will see that the tube is facing you in the control panel on HTP view. You can't lift the table, you cannot lift it up or down, so I think that's another challenge with the machine".*

*[Radiographer 10]*

During observations it was noted that all four hospitals had no trolleys which allows examinations like x-ray of the pelvis, abdomen and femur without needing to transfer patients to the x-ray table. These are particularly necessary for patients with fractures coming from the Emergency Department, who ideally must have their X-rays whilst on the trolley with very minimum movement. Trolleys that the researcher saw were not x-ray compatible and had no trays underneath for placing cassettes or image receptors.

Finally, the observations showed that essential consumables such as contrast media, films, gloves and syringes were in shortage in all the three public hospitals. During one observation, a patient was told to go to another hospital or privately to have CT angiogram. In one of the public hospitals, no contrast pump was available to perform contrast media examinations. At another public hospital, the pump was not working for all most two months. In both interviews and focus groups participants mentioned that they are unable to offer optimum service due to the shortage of consumables:

*"I think shortages of consumables like films and currently in departments we are left with about 100 films the whole department, so it also affects us in providing optimum services and it also in contrast media sometimes we operate without contrast media. The patient had to go out and look for their own contrast media and come for the examination which is of a problem in providing radiology services"*

[Radiographer 11]

*"As I am doing this interview there are no films for the CT scan. They are just using discs, which are good, but some people prefer some hard copies like films you know".*

*[Participant Snr01]*

The effects equipment shortages were observed in the delivery of patient care. During field observations, LT observed patients at the three public hospitals moved from the bed or trolly to

the X-ray table or CT scanner inappropriately. For example, on one occasion, the patient was pulled across by the bedsheet while screaming in pain. On a different day at the same hospital, a child was lifted onto the X-ray table while on traction for a femur X-ray. Furthermore, during observations, LT noted that sometimes the radiographers do not close the doors during radiation exposure—people walked in and out of the controlled room despite the warning light being on. X-rays and CT scans were performed with relatives or carers standing in the room unprotected.

### (ii) Suboptimal equipment repair and maintenance

In addition to the scarcity of available equipment and infrastructure for radiology services, we found common problems arising with maintaining equipment. Interviewees and focus group participants expressed concerns for the lack of support and resources from hospital management for equipment repair. At the three public hospitals there were machines lying idle, including a portable X-ray machine meant to be in children's clinic and critical care unit that had not been repaired in two years. Only two of these hospitals had portable machines in working order. In these three hospitals some X-ray rooms were not in use because the machines had broken down. According to the focus groups participants, they had broken down due to machine aging, lack of servicing/repair, overuse and erratic electricity supplies. Unreliable air conditioning for cooling had also contributed to machine overheating. Overall, however, they felt that the lack of machine servicing was the main cause for the breakdowns:

> *"We rarely service our machine because I* [*have*] *never seen a machine working and someone just come in and say* [*he*] *is here to service the machine. I never seen this in a government hospital*
>
> *[Radiographer 7]*

Participants reported that none of the hospitals had a maintenance department whom they inform when machines break down. However, they reported that repair engineers did not have the required skills nor competencies for the repair requirements and needed to source repairs abroad from South Africa or Germany, which was not easy to finance, and resulted in further delays in servicing.

> *"The hospital equipment and maintenance technicians–- they haven't been trained to these current modalities how to repair the modern machines, so the skewed gap has affected the whole area of the hospital. . . we have got to outsource the services from outside the hospital, which is very expensive for the hospital"*
>
> *[Radiology manager 2]*

As a result of these maintenance shortfalls, there are concerns for patient safety as there are few quality assurance checks for the machines:

> *"We are lacking very much there if there is any quality assurance at all that is done.* [*Quality Assurance*] *testing of machine is very rare and no proper record of it stored"*
>
> *[Radiographer 7]*

In public hospitals it was noted that radiological equipment was installed without proper training of staff on appropriate use. One of the participants strongly stressed that there is no point of having a new machine when radiographers are not trained.

*"As much as we are qualified, we have working experience we still need training to catch up with ongoing advancements in radiology services for example say we get a new machine. We need training . . . to optimally use the machine. . . with all the new equipment that we get we learn by trial and error at the end we don't use that machine optimally because we are not aware of how much it can do".*

[Radiographer 7]

### (iii) Shortage of radiology staff and skills development

One major cause of concern from participants is the shortage of radiology staff, resulting in working out of hours and fatigue/exhaustion and sometimes absences.

*"You find that at times you are forced to book patients on appointment for next day, like in ultrasound and CT, you have to book some patients because there are not enough radiographers on the ground . . . there is a shortage of staff, the number is not enough".*

[Senior Radiographer 1]

*"The problem we have here is that we are short-staffed. The major problem is that you cover night then you must come back to work the following day. . . if our staff is slightly increased such that after you cover night you [can] take a day off which is so healthy".*

[Radiographer 5]

As a result of staff shortages, students were often asked to fill in the service gaps, but the observations at hospitals A, B and C showed that they were unsupervised.

*"I don't think we have enough radiographers because in other modalities, I have seen that they have to be done by a student sometimes because we won't have enough time on standby, so I think we don't have enough radiographers.*

[Radiologist 1]

During observations at one of the public hospitals, LT witnessed two young boys undergoing femur X-rays with no gonadal shields used, exposing them to potentially harmful radiation. The X-rays were performed by radiology students who were not supervised at the time. They also failed to identify the patients correctly due to an incorrect form and had ended up X-raying the wrong patient, although this was rectified during the procedure and the image was edited and assigned to the correct patient before printing.

The three public hospitals did not have a resident radiologist, which was highlighted by the participants as one of the main reasons why optimal radiology services are difficult to provide. One public hospital has a radiologist who covers for less than four hours in the morning. Shortage leads to late reports issued after two or more days whilst waiting to confirm reports from radiologists in the private sector and start treatment. All the participants agreed that the shortage of radiologists do affect the whole chain of care by not having reports on time and much needed guidance from radiologists.

Consequently, reporting of images is very poor as it takes days for reports to be out.

*"Some of the examination that are done are not reported then the clinicians would then report the examinations. . . wouldn't say they are optimum work since it must be done by a*

*radiologist. The one who is reporting for the service is not trained, so they are a lot of misdiagnosis which leads to mismanagement of patients.*

*[Radiologist 1]*

According to the participants, patients may need to pay more to expedite the reporting of their investigation by the radiologist. It was noted that some patients do not come back with a report or they will come two or more days later with the radiologist's report.

### (iv) Lack of wider integration and support for radiology services

Our final overarching identified theme is the lack of support more widely for radiology services from the health system, including other health providers and state support. This was manifested in numerous ways, starting with clinician referrals: we observed that radiology referral request forms were inadequately filled in by clinicians, and not following referral request standards (see Table 1). Request forms often lacked patient demographics, the required examination on x-ray, the name of the referring clinician, dates and/or signatures. The participants stated that it is an on-going issue that needs to be rectified for them to properly carry out the appropriate examination. At the end patients suffered due to lack of vital information required to be on the form as they may end up with repeated un-necessary x-rays or were sent back to the referrer to adequately fill the form.

*"Sometimes you send back the patient for the physician or for the referring doctor to sign and give us an indication on a procedure and sometimes the patient just comes back, the area is still blank, and in the end you just end up offering the service generally without knowing exactly the indication, what the doctor expect and after that you do your procedure and you give them images. They come back saying that's the wrong thing that we have done and x-ray the patient again. . . We don't have to put the patient in between us, sending the patient back and forth and in the end the patient suffers*

*[Radiographer 4]*

LT noted that many of the referral forms were poorly written, and that it could be difficult for radiographers to obtain clarity and corrections in the forms faced with uncooperative and sometimes harassing behaviour from clinicians. LT observed one doctor shouting at the radiology staff of one of the public hospitals for questioning the form, even though inadequate information was on the form to enable the radiological examination to be made or for the patient to be identified correctly.

In terms of financial support, for public hospitals there can be little space for improvements on some fronts, since funding for infrastructure, equipment and consumables tends to come from state funding. Interviewees noted, for example, that while they may have some equipment available, the running costs of consumables, for example, could not be maintained by the hospital's funds.

There are also not many opportunities for career development for radiologists and radiographers. Participants expressed their desire to further their studies and skills in radiography but stated they did not have access to existing opportunities abroad. However, they pointed out that even if there were training opportunities, there are no allocated government funding arrangements for training. Similarly, radiology training does not exist in Zimbabwe; and finding funding or sponsorship to train abroad (South Africa, Uganda or the UK) is difficult. Many who do self-fund for training abroad do not return due to better work opportunities abroad.

*"They are few people training to be radiologists because they still need sponsorship to get outside to be trained but the few that are trained outside don't want to come back because they have found that outside is greener than Zimbabwe at the moment because we have few Zimbabweans who are radiologists, who are working in Australia, UK and South Africa.*

[Radiologist 2]

*"Sometimes it is just an in-house training like ultra sound in this department where senior radiographer and the head of the department teach everybody how to do an ultrasound... at university they have recently started master's level I know".*

[Radiographer 1]

### (v) Strong motivation as a facilitator for improvement

Despite the numerous infrastructure, equipment, staff and financial support challenges identified so far, we identified a strong indication of high levels of motivation among staff to improve radiology services and improve patient care. This was reflected in the language adopted by the participants in describing their job, including phrases such as 'I am involved in motivating colleagues into team work', we 'make things happen' and 'we go the extra mile'. Participants particularly acknowledged the Iifficulties presented by working in a low-resource setting, and 'going the extra mile' in order to compensate for this was a recurring theme in the interviews:

*"We do with the limited resources that we have. We go an extra mile to use the few resources to produce the best outcomes"*

(Radiology manager 2)

*"We have tried under very difficult circumstances to remain functional and to save our general public"*

(Radiology manager 4)

The notion of 'going the extra mile' was couched by some in terms of moral duty, citing service of the public, and noble acts as motivating factors:

*"You know every day you wake up and the most important thing you just want to do is good things to the patients, but now at times you face some challenges as you do your work in terms of resources, constrains... It might take time, but we always thrive to serve the patients"*

(Senior radiographer 1)

*"If those radiologists in the private sector can afford to give a service to the public sector, I think it should be recorded as a noble thing"*

(Radiology manager 4)

When asked what their view of the good work that they do was, one participant was keen to draw out the importance of maintaining the service for the community, rather than focus on the work of individuals:

*"The good things I do. Actually I see this as an ongoing concern... It should be something that goes on and on, even if I leave the place."*

*(Senior radiographer 2)*

One manager (head of department) commented that the gap between remuneration in public and private settings has an impact on staff motivation:

*"Working conditions in the public sector are not as favourable in terms of remuneration and staff. So, the staff is not fully motivated hence the service delivery is lacking, but I hope if the economy improves remuneration and working conditions also improves for the public sector because the public sector should actually be the backbone of any country's health delivery system."*

*(Radiology manager 3)*

Similarly, managers warned that lack of staff could lead to overwork and burnout:

*"You don't want to be on call every time, you will burn out and you become too exhausted to want to come to work"*

*(Radiology manager 1)*

We found these comments across the transcripts from all four hospitals, particularly from the senior staff in the interviews. Comments from more junior staff (radiographers) in the focus groups discussions also suggest this sense of duty and motivation, including raising the awareness of radiology as an important service in the hospital and the desire for further development and training to better perform in their jobs.

## Discussion

In this study we have identified barriers for delivering radiology services (in Zimbabwe) derived from interviews and focus groups, and validated via observations, from the perspective of radiology staff of varying levels of seniority in three public and one private hospital. In this section, we first revisit our findings and relate them to the standards set for safe delivery of radiology services to discuss why these identified barriers impede quality radiology service delivery. We then discuss the potential facilitators for improving services by emphasising the need for building on the existing motivation for delivering care that we identified in the Zimbabwean context.

### Relating identified barriers to quality and safety of radiology delivery

While we stress that our study was not intended to be a point-by-point comparison or audit of safety standards in radiology services and current practices, we propose that the barriers to radiology service delivery reported by the participants in our study do point to risks to patients and staff in the delivery of radiology services. Starting with basic infrastructure, equipment and consumables, we observed that the current state of radiology buildings and equipment fall short of the recommended guidelines by the WHO and other standards (see Table A1 in S1 Appendix) and illustrate the scarcity of radiology resources in Zimbabwe found in the audit by Maboreke et al. (2019) [3] and are similar to those found in other lower-income settings (e.g. Tanzania [4]). The inadequate referral forms we observed mirror experiences in Ghana [5] and Nigeria [6, 28]. These basic requirements, alongside staff shortages, harassment and limited training reported by our data, have implications for patient safety within radiology services as pointed out by previous literature [22]. Furthermore, we saw no evidence of a protocol

or system in place in which potential incidents could be reported nor recorded or monitored, which are the basic requirements for addressing patient safety concerns and adopting a 'safety culture' [29, 30]. Taken together, the safety risks posed by these physical infrastructure and human resource challenges are therefore crucial barriers to the safe delivery of radiology services and are therefore crucial barriers.

### Reflecting on facilitators for improvement of radiology services

Reflecting on our observations and the latest literature concerning improvement on patient safety and a safety culture, we are conscious that studies examining patient safety and/or quality improvement efforts across healthcare systems highlight the importance of staff motivations, leadership and behaviour. For example, a recent review of improvement efforts in healthcare noted that while there is no 'magic bullet' to quality improvement interventions, these are much more likely to succeed "when they are developed with, rather than imposed on, healthcare professions" and when "fostering a sense of ownership" [31]. Moreover, they note that organisational cultures "supportive of personal and professional development, and committed to improvement as an organisational priority" are more likely to "provide an environment where improvement efforts can flourish" [31, 32] Against this background, we note that the strong sense of personal motivation and willingness to improve services observed in our study is crucial for identifying opportunities for improvement of radiology services, and present a conduit towards facilitating improved radiology services. Participants described the need to improve services in moral terms, referring to public service and noble acts. Our study did not identify the reason for this sense of motivation despite the resource limitations of the Zimbabwean context, which would merit further investigation. However, we argue that these positive sentiments and motivations we found are unlikely to last if there are continuous challenges and pressure on staff to deliver: we observed some signs of fatigue, exhaustion, and harassment, in addition to low public sector pay, some of which has been observed in other radiology studies [33]. The potential for 'brain drain' in Zimbabwean health service staff has been highlighted in a review by Chibango (2013) [34], and of African continent in general [35] and this would need to be avoided [34]. One way to avoid losing more staff and/or demotivating them is to invest in training. Some work has been taking place in other countries: a review of radiology training needs in rural settings in Uganda, Kenya, Tanzania, Rwanda, Zambia, Ghana, Malawi, and Sudan found that there had been national state support for implementing training interventions, provided by public, private-for-profit), private-not-for profit, local, and international academic institutions, personal initiatives, and professional societies [36]. However, there were still gross disparities in rural services despite these efforts, calling for more focussed and organised investments in training [36]. We therefore argue that as a way of facilitating improved radiology services in the Zimbabwean context, there is a need to prioritise investing in training and retaining of radiology staff in the country alongside the investments for basic infrastructure.

The opportunities for facilitating radiology service improvement in terms of infrastructure and equipment have been highlighted by others and will not be repeated here [2, 3]. However, we do note that there wider organisational challenges to overcome which would facilitate the context in which radiology services can be delivered. For example, the challenges in the way radiology is acknowledged by clinicians operating in other areas of the healthcare system we identified requires management and organisational changes. The lack of servicing and training may imply that procurement mechanisms need to be improved to allow for service and maintenance contracts that cannot be currently provided in-house [37]. As with any service delivery

improvement required in hospitals globally, infrastructure changes would have to go hand-in-hand with organisational changes so that these can be sustained over time.

## Limitations

Although appropriate for the qualitative study design, we note the small number of participants in our study, and that it was limited to four hospitals in the capital region. Based on the 2019 audit of radiology equipment units in the country [3], we know that Harare is one of the two cities in which most of the registered radiology equipment units are operating for the whole country. This means that the four included hospitals represent a significant proportion of radiology services available for the region, and implies that quality of services in less affluent public hospitals is likely to be lower. However, we do acknowledge that the lack of data from the second biggest city provider of radiology services (Bulawayo), does not provide a full picture of service provision in the country. Furthermore, we note that separate focus groups were not conducted with the senior participants included in the interview sample (managers, senior radiographers, radiologists and consultant physicians), due to resource constraints of conducting fieldwork in the Harare setting. Our findings are therefore likely to be more representative of the less senior radiographers. We also did not include perspectives of radiation authorities or other decision-makers who could act upon the findings of this study. We felt these individuals were out of scope as our focus was on those delivering the services on the ground and identifying their perspectives, but propose the inclusion of these wider stakeholders in future studies would facilitate the steps towards finding solutions for improving radiology services. On our data collection methods, we note that observations only took place for 1–2 days, and only in the three public hospitals. Ideally, there would have been observations in all four hospitals over a longer period covering different days of the week, given that activities and functions may vary across a given time. Given that these observations served to validate contextual information given by the interviewees and focus group participants, and not provide new data and insights, this limitation is partly mitigated.

## Conclusions

The main barriers to delivering radiology services we identified were poor basic equipment, infrastructure and consumables; sub-optimal equipment use and maintenance; a shortage of radiology staff; and lack of wider integration and support for radiology services. These findings point to potential risks to patient safety and quality of delivering radiology services. However, the strong sense of personal motivation displayed by the staff in this study suggests there is the potential to maintain and improve existing practices. We view this potential as a facilitator to how future improvements of radiology services, but this would require investments to train and remunerate more radiology staff, and investing in continuing professional development, alongside the basic equipment and maintenance investments.

## Supporting information

**S1 Appendix. Categories of factors affecting radiology services identified from published literature, guidelines and standards.**
(DOCX)

**S2 Appendix. Interview protocol.**
(DOCX)

**S1 Checklist. Inclusivity checklist.**
(DOCX)

**S2 Checklist. SRQR checklist.**
(DOCX)

## Acknowledgments

The authors would like to thank all the participants of the interviews and focus groups who gave up their time for this study. We are particularly grateful to the managers of the four hospitals in the study for facilitating access to the staff.

## Author Contributions

**Conceptualization:** Lazarus Tombo, Edith D. Majonga, Carole Cummins, Semira Manaseki-Holland.

**Data curation:** Lazarus Tombo.

**Formal analysis:** Lazarus Tombo, Semira Manaseki-Holland.

**Investigation:** Lazarus Tombo, Carole Cummins, Semira Manaseki-Holland.

**Methodology:** Lazarus Tombo, Carole Cummins, Semira Manaseki-Holland.

**Project administration:** Lazarus Tombo, Edith D. Majonga, Carole Cummins, Semira Manaseki-Holland.

**Resources:** Edith D. Majonga.

**Validation:** Saba Hinrichs-Krapels, Lazarus Tombo, Harriet Boulding, Edith D. Majonga, Carole Cummins.

**Writing – original draft:** Saba Hinrichs-Krapels, Lazarus Tombo, Harriet Boulding, Semira Manaseki-Holland.

**Writing – review & editing:** Saba Hinrichs-Krapels, Lazarus Tombo, Harriet Boulding, Edith D. Majonga, Carole Cummins, Semira Manaseki-Holland.

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
