## [Decision Letter · Decision Letter 0]

27 Jul 2022

PGPH-D-22-00502

Barriers and facilitators to the provision of radiology services in Zimbabwe: a qualitative study

Dear Dr. Hinrichs-Krapels,

Thank you for submitting your manuscript to PLOS Global Public Health. After careful consideration, we feel that it has merit but does not fully meet PLOS Global Public Health’s publication criteria as it currently stands. Therefore, we invite you to submit a revised version of the manuscript that addresses the points raised during the review process.

When revising your manuscript, please address reviewer 1's comments in terms of clarifying the study's justification as well as the methods. As part of these clarifications, it would be helpful to go into some more detail about the Zimbabwean healthcare system, as mentioned by reviewer 2.  We would also ask that you reframe the work's findings more clearly, as well as its conclusions. Please see the reviewers' comments below. This is a very interesting and important article, and I look forward to reading the revised version.

We look forward to receiving your revised manuscript.

Kind regards,

Anat Rosenthal

Academic Editor

Journal Requirements:

2. Please add a full list of legends for all your Supporting Information files after the references list.

Reviewers' comments:

Reviewer's Responses to Questions

**Comments to the Author**

1. Does this manuscript meet PLOS Global Public Health’s publication criteria? Is the manuscript technically sound, and do the data support the conclusions? The manuscript must describe methodologically and ethically rigorous research with conclusions that are appropriately drawn based on the data presented.

Reviewer #1: No

Reviewer #2: Yes

2. Has the statistical analysis been performed appropriately and rigorously?

Reviewer #1: No

Reviewer #2: No

3. Have the authors made all data underlying the findings in their manuscript fully available (please refer to the Data Availability Statement at the start of the manuscript PDF file)?

Reviewer #1: Yes

Reviewer #2: Yes

4. Is the manuscript presented in an intelligible fashion and written in standard English?

Reviewer #1: Yes

Reviewer #2: Yes

5. Review Comments to the Author

Reviewer #1: The title

Since the paper deals with experiences and perceptions of radiology human resource, the title “Barriers and facilitators to the provision of radiology services in Zimbabwe: a qualitative study” is too generic and does not explicitly focus on the aim of the study. This study sought out to explore the experiences and perceptions of radiology workers which impact provision of quality radiology services in Zimbabwe. and; a suggested title is “Radiology professionals’ perceptions and experiences affecting delivery radiology services in Zimbabwe” or “Experiences and perceptions of radiology workers that affect provision of quality radiology services in Zimbabwe”.

Justification

There is need for a paragraph to justify the study i.e; Radiology services in LMIC are inadequate to meet the health needs of these countries. Radiology is critical for accurate disease diagnosis, screening and image-guided therapies. Imaging services in low and middle income countries have received little attention. Most research has analyzed physical infrastructure and equipment, however, experiences and perceptions of service providers is critical to effective service provision. This area has not yet been researched on and presents a significant knowledge gap. There is therefore a justification for exploring the experiences and perceptions of radiology workers as they engage in service delivery, and explain how this affects quality service provision.

Methodology

Choice of Zimbabwe as a setting is logical since it is presumed to be representative of some other radiology departments in low resource settings especially Africa.

An exploratory qualitative study design with triangulation of data sources; observation, interview and focus group discussions is appropriate for investigating experiences and perceptions. Although the researchers imply triangulation Line 141, they don’t explicitly state how triangulation was conducted, which research question was answered by triangulation and the type of triangulation used.

The study aimed at achieving rigor through triangulation of methodologies and data sources. Ideally for triangulation to be effective, the different methodologies namely interview, focus group discussions and observations must all address the objectives and explore the same variables. In this case, the variables were experience, perceptions and attitudes. It is apparent that the observation methodology did not focus at experience, perceptions and attitudes.

Furthermore, an attempt should be made to ensure the target groups are the same for each method of triangulation. As an example, radiographers should feature in the interviews, FGDs and they should be the target for the observation component. Similarly, radiologists should feature in the interviews, FGDs and they should be the target for the observation component. Managers should also feature in the three methods of data acquisition.

The uniform involvement of all types of participants in the three methodologies, in addition to them all addressing the same issues, ensures data homogeneity, comprehensiveness and saturation and allows for comparison of responses and gleaning of data for concepts, subthemes and themes. This study does not ably depict that homogeneity.

Table 1 should be in an annex or just referred to, but should not form the body of this paper, since it is not a main finding but it is just a reference standard.

Lines 117 to 119 are confusing. The numbers don’t tally. Does the sample comprise of a total of 4 (three public and one private) or a total of 6 (four public and two private) hospitals?

Why were only 13 staff interviewed? Did they interview until the responses were saturated? How was saturation of the answers ensured?

What governed the choice of the carder who participated in the interviews and those who participated in the focus group discussions? Why were the managers, senior radiographers, radiologists and consultant physician not included in the focus group discussions? If as the authors state that the exclusion of higher carder was to remove un-due influence, a focus group discussion composed of these higher carders only should have been conducted.

There is an inconsistency in the number of hospitals where observation was conducted. The test states two hospitals yet the table shows three. Why were observations only conducted in two of the facilities and what factors guided the selection of these two facilities? One day is may not be sufficient to conduct an observation study since activities and functions may vary through an entire week and month. The observation based on just one day may therefore be under-representative of the real situation.

The field observations should have focused at the same objectives namely attitude, experiences and perceptions to assess how these were lived out and evident in the day to day practice, instead, the observers focused on physical infrastructure, buildings and equipment. Observations should have targeted staff and their behavior. The information from these observations can therefore not be triangulated in the analysis since the focus is entirely different and not in line with the study aims.

Findings and discussion

In the results, the researchers need to structure their lay out in relation to their objectives i.e., we need to see results relating to experiences and perceptions. They should also these results by data collection technique as stated in the methodology.

Additionally, some of major findings don’t however fall in the main realm of experiences and perceptions and don’t address all objectives. The authors list 2 objectives namely:(a) describe staff experiences and perceptions in delivering., (b) identify staff barriers for delivering radiology services. In addition, these objectives and results don’t align well with the study title. Question is; are the experiences also the facilitators and barriers? There is need for clarity of what the researchers set out to do.

The poor basic infrastructure, lack of equipment, and consumables, suboptimal equipment maintenance; and integration of radiology services into other hospital functions are neither an experiences or a perceptions and don’t have a direct linkage to staff. They could have been reported on but they should not be at the forefront of the findings and discussion since the paper deals with radiology workers and their “experiences and perceptions”.

The “strong sense of motivation” would fall within the subject of experiences and perception and should have been the main area of discussion. It is not mentioned however that this motivation was universal across all carders and facilities and whether there are some facilities or carders who were not strongly motivated. The explanation to this strong sense of motivation is also not given.

The shortage of radiology staff and skills development justifiably falls within the realm of the research but there is need to explore and get the views of the staff on the optimal number of professionals and their ranks and areas of specialization, plus the anticipated areas of skills training.

The authors should reorganize the results and discussion to focus more at the experiences, perception and attitude which impact radiology service provision, rather than the physical aspects of infrastructure and equipment since these are covered in other publications (as indicated in the introduction), and the justification of this research as well as the objects were to explore how these experiences, perception and attitude impact radiology services. The authors may then propose ways of mitigating the impact of these on service provision.

The researchers focus not on their study results in their discussion. They need to discussion why their results turned out the way they did and compare with the existing body of knowledge. The discussion section should also be structured according to the results based on the study objectives. See line 463

Conclusion

We don’t see any conclusions relating to the perceptions.

Reviewer #2: A good study whose findings add to a knowledge gap, and point to key issues within the radiology services provision in the country, and may stimulate relevant actions towards the infrastructural, equipment, skills and services required to optimize the services. The title is appropriate for the study aims, methodology and conclusions.

The statement of intent is clear, the methods are well articulated with use of validation and benchmarking to bridge the local gaps, and this makes for a comprehensive picture. Though qualitative the authors needed to include some background quantitative data on workload statistics and specifics on burden of disease for which radiology is required in order to adequately put into perspective the deficit in services provision.

The lack of input from the respective hospitals managers, the radiation protection authority, and Ministry of Health who have the capacity to act on the findings should be addressed. At least for the hospitals A to D the key findings should have been tabled for specific improvements, and clearer comparisons drawn. A limitation is lack of data from the second city, Bulawayo, and other cities which offer radiology services. This further limits the conclusions on the specifics of equitable access to the services and what spectrum of services is available in other locations in the country. The authors must therefore declare the latter as a weakness of their study to open for further study. The authors point to the challenges in equipment purchase and they should make sound recommendation from their findings, such as service contracts which take into account serviceability by the equipment suppliers

6. PLOS authors have the option to publish the peer review history of their article (what does this mean?). If published, this will include your full peer review and any attached files.

**Do you want your identity to be public for this peer review?** For information about this choice, including consent withdrawal, please see our Privacy Policy.

Reviewer #1: No

Reviewer #2: **Yes: **DR PORTIA MANANGAZIRA

---

## [Decision Letter · Decision Letter 1]

15 Mar 2023

Barriers and facilitators for the provision of radiology services in Zimbabwe: a qualitative study based on staff experiences and observations

PGPH-D-22-00502R1

Dear Dr Hinrichs-Krapels,

We are pleased to inform you that your manuscript 'Barriers and facilitators for the provision of radiology services in Zimbabwe: a qualitative study based on staff experiences and observations' has been provisionally accepted for publication in PLOS Global Public Health.

Best regards,

Julia Robinson

Executive Editor

Reviewer Comments (if any, and for reference):

Reviewer's Responses to Questions

**Comments to the Author**

1. If the authors have adequately addressed your comments raised in a previous round of review and you feel that this manuscript is now acceptable for publication, you may indicate that here to bypass the “Comments to the Author” section, enter your conflict of interest statement in the “Confidential to Editor” section, and submit your "Accept" recommendation.

Reviewer #3: All comments have been addressed

2. Does this manuscript meet PLOS Global Public Health’s publication criteria? Is the manuscript technically sound, and do the data support the conclusions? The manuscript must describe methodologically and ethically rigorous research with conclusions that are appropriately drawn based on the data presented.

Reviewer #3: Yes

3. Has the statistical analysis been performed appropriately and rigorously?

Reviewer #3: Yes

4. Have the authors made all data underlying the findings in their manuscript fully available (please refer to the Data Availability Statement at the start of the manuscript PDF file)?

Reviewer #3: Yes

5. Is the manuscript presented in an intelligible fashion and written in standard English?

Reviewer #3: Yes

6. Review Comments to the Author

Reviewer #3: No further comments. All concerns were addressed appropriately.

7. PLOS authors have the option to publish the peer review history of their article (what does this mean?). If published, this will include your full peer review and any attached files.

**Do you want your identity to be public for this peer review?** For information about this choice, including consent withdrawal, please see our Privacy Policy.

Reviewer #3: No
